# International Consensus on Definition of Mild-to-Moderate Ulcerative Colitis Disease Activity in Adult Patients

**DOI:** 10.3390/medicina59010183

**Published:** 2023-01-16

**Authors:** Bénédicte Caron, Vipul Jairath, Ferdinando D’Amico, Sameer Al Awadhi, Axel Dignass, Ailsa L. Hart, Taku Kobayashi, Paulo Gustavo Kotze, Fernando Magro, Britta Siegmund, Kristine Paridaens, Silvio Danese, Laurent Peyrin-Biroulet

**Affiliations:** 1Department of Gastroenterology and Inserm NGERE U1256, Nancy University Hospital, University of Lorraine, 54500 Vandoeuvre-lès-Nancy, France; 2Department of Medicine, Western University, London, ON N6A 3K7, Canada; 3Department of Epidemiology and Biostatistics, Western University, London, ON N6A 3K7, Canada; 4Gastroenterology and Endoscopy, IRCCS Ospedale San Raffaele and University Vita-Salute San Raffaele, 20132 Milan, Italy; 5Department of Biomedical Sciences, Humanitas University, Pieve Emanuele, 20090 Milan, Italy; 6Gastroenterology Division, Rashid Hospital, Dubai Health Authority, Dubai 003206, United Arab Emirates; 7Department of Medicine I, Agaplesion Markus Hospital, Goethe-University, 60431 Frankfurt am Main, Germany; 8Inflammatory Bowel Disease Unit, St. Mark’s Hospital, Harrow HA1 3UJ, UK; 9Center for Advanced IBD Research and Treatment, Kitasato University Kitasato Institute Hospital, Tokyo 108-8642, Japan; 10IBD Outpatient Clinics, Colorectal Surgery Unit, Pontificia Universidade Católica do Paraná (PUCPR), Curitiba 80215-901, Brazil; 11Unit of Pharmacology and Therapeutics, Department of Biomedicine, Faculty of Medicine, University of Porto, 4200319 Porto, Portugal; 12Department of Clinical Pharmacology, São João University Hospital Center (CHUSJ), 4200319 Porto, Portugal; 13Center for Health Technology and Services Research (CINTESIS), 4200319 Porto, Portugal; 14Charité–Universitätsmedizin Berlin, Corporate Member of Freie Universität Berlin, Humboldt-Universität zu Berlin and Berlin Institute of Health, 10117 Berlin, Germany; 15Department of Gastroenterology, Rheumatology and Infectious Disease, Campus Benjamin Franklin, 10117 Berlin, Germany; 16Ferring International Center S.A. Ch. De la Vergognausaz 50, 1162 Saint-Prex, Switzerland

**Keywords:** ulcerative colitis, mild, moderate, definition, activity

## Abstract

*Background and Objectives*: At present, there is no consensus definition of mild-to-moderate disease activity in patients with ulcerative colitis. The objective of the present study was to establish a reliable definition of mild-to-moderate disease activity in adult patients with ulcerative colitis. *Materials and Methods*: Twelve physicians from around the world participated in a virtual consensus meeting on 26 September 2022. All the physicians had expertise in the diagnosis and treatment of inflammatory bowel disease. After a systematic review of the literature and expert opinion, a modified version of the RAND/University of California, Los Angeles appropriateness method was applied. A total of 49 statements were identified and then anonymously rated (on a 9-point scale) as being appropriate (scores of 7 to 9), uncertain (4 to 6) or inappropriate (1 to 3). The survey results were reviewed and amended before a second round of voting. *Results*: Symptom and endoscopic-based measurements are of prime importance for assessing mild-to-moderate ulcerative colitis activity in clinical trials. The experts considered that clinical activity should be assessed in terms of stool frequency, rectal bleeding and fecal urgency, whereas endoscopic activity should be evaluated with regard to the vascular pattern, bleeding, erosions and ulcers. Fecal calprotectin was considered to be a suitable disease activity marker in mild-to-moderate ulcerative colitis. Lastly, mild-to-moderate ulcerative colitis should not have more than a small impact on the patient’s daily activities. *Conclusions*: The present recommendations constitute a standardized framework for defining mild-to-moderate disease activity in clinical trials in the field of ulcerative colitis.

## 1. Introduction

In terms of severity, ulcerative colitis (UC) is typically classified as being “mild-to-moderate” or “moderate-to-severe” [1]. However, a great variety of definition of “mild-to-moderate” disease activity in UC can be found in the medical literature and in clinical practice [1]. Several metrics have been developed to monitor and standardize the assessment of clinical activity in UC; these include the Simple Colitis Clinical Activity Index (SCCAI), the Mayo Clinic Score (MCS), the Ulcerative Colitis Disease Activity Index (UCDAI) and the Truelove and Witts criteria for severe disease [2,3,4,5,6,7]. Nevertheless, there is no consensus on the definition of mild-to-moderate disease activity in UC [1,8,9].

Sedano et al. have systematically reviewed definitions of mild-to-moderate UC found in protocols listed at clinicaltrials.gov (accessed on 10 October 2022) [10]. The MCS was the most frequently used score, while the UCDAI was detected in a small proportion of trials (13.1%) [10]. Twenty different MCS cut-offs have been used to define mild-to-moderate active UC: The minimum cut-off ranged from 1 to 6, and the maximum cut-off ranged from 4 to 11 [10]. However, the MCS and UCDAI have some limitations because they include the subjective Physician Global Assessment (PGA) sub-score [11]. Most regulatory authorities recommend excluding the PGA sub-score in order to reduce subjectivity and thus focus on the patient’s self-reported symptoms and objective endoscopic findings [11].

Sedano et al.’s review emphasized that the definitions of mild-to-moderate UC vary markedly from one clinical trial to another. The lack of a consensus on the definition of mild-to-moderate UC means that the clinical trial data are heterogeneous and non-reproducible. We therefore lack a standardized definition of mild-to-moderate UC disease activity in patients eligible for inclusion in clinical trials; this constitutes a key unmet need in the field of inflammatory bowel disease (IBD).

In the present study, we first comprehensively reviewed the literature on (i) definitions of mild-to-moderate active UC and (ii) factors that were predictive of a treatment response in randomized controlled trials (RCTs) of 5-aminosalicylic acid (5-ASA) and/or budesonide Multi Matrix^®^. Secondly, we formed an international expert panel and conducted a multiple-round survey. The objective was to establish a robust definition of mild-to-moderate UC disease activity for use in clinical trials with adult patients.

## 2. Materials and Methods

Firstly, we systematically reviewed the definitions of mild-to-moderate active UC used in 39 RCTs of 5-ASA and/or budesonide Multi Matrix^®^ [12]. Six different indexes were used to define mild-to-moderate active UC in these trials—emphasizing the high degree of heterogeneity in the literature [12]. Most RCTs used the UCDAI [12]. Four different UCDAI cut-offs were used to define mild-to-moderate active UC. The most common UCDAI cut-offs (reported in more than half of the included RCTs) were ≥4 and ≤10, with a sigmoidoscopy score of ≥1 and a PGA score ≤2 [12].

Secondly, we used the modified RAND/University of California, Los Angeles (UCLA) appropriateness method by incorporating a Delphi panel approach with iterative rounds of voting and discussion [13]. This approach combined the best available evidence with expert opinion, in order to (i) assess the face validity and the feasibility of items identified in the systematic review and (ii) generate a robust definition of mild-to-moderate disease activity in UC [12,13].

There were two rounds of voting. Two of the authors (VJ and BC) prepared 49 preliminary statements based on the needs identified in the systematic review. The list of statements was then disseminated online. The expert panel members anonymously rated each item for appropriateness on a 9-point scale (ranging from 1 = inappropriate to 5 = uncertain and 9 = highly appropriate). As specified in the RAND/University of California Los Angeles (UCLA) manual, each statement was classified (according to the panel’s median rating and extent of disagreement) as inappropriate (a median score of 1 to 3.5, with no disagreement); uncertain (a median score of 3.5 to 6.5 with no disagreement or a median score of any value with disagreement); or appropriate (a median score of 6.5 to 9, with no disagreement). Disagreement for a given statement was defined as six or more votes in the lowest three-point region (i.e., 1–3) and six or more votes in the highest three-point region (i.e., 6–9).

The first-round survey results were reviewed, discussed and amended during a videoconference that took place on 26 September 2022. The latter included 12 physicians from nine different countries (Brazil, Canada, France, Germany, Italy, Japan, Portugal, the United Arab Emirates and the United Kingdom); all had significant expertise in the field of IBD. The videoconference’s objective was to identify areas of disagreement on item appropriateness and the rationale for answers. The survey was then revised as a function of the panel’s discussions in order to clarify statements prior to a second round of voting. This second round occurred only if no agreement was achieved during the first round, and the appropriateness of statements was scored in the same way. If no agreement was found, the statement in question was excluded. All the experts helped to write the manuscript and approved the final version for publication.

## 3. Results

The results of our systematic review of the literature (39 RCTs) have been published elsewhere; they emphasized the great variety of definitions of mild-to-moderate UC applied to the inclusion of patients in RCTs [12].

### 3.1. Item Generation and the Survey

The previously published results of the systematic review were used to support the survey statements. Items were grouped according to the following topics: symptom-based disease activity assessments, endoscopy-based disease activity assessments, histology-based disease activity assessments, biomarker-based disease activity assessments, composite disease activity scales, societal guideline-based definitions of mild-to-moderate UC and quality of life/disability-based definitions of mild-to-moderate UC. The first survey consisted of 49 items. The virtual panel for the final survey comprised 12 voting members. Overall, 29 (59.2%) items were considered to be appropriate. Three (6.1%) items were discussed, voted on and approved after the second round of voting. The statements on which the experts agreed are summarized in Table 1. Statements that were excluded are shown in Appendix A.

### 3.2. Symptom Based Disease Activity Assessments

The panel decided that measurements of symptoms were important for defining mild-to-moderate UC. The Clinical Activity Index (CAI), the Disease Activity Index (DAI), the UCDAI and the MCS have been used in clinical trials. The panel considered that the UCDAI or the MCS should be used to assess mild-to-moderate UC disease activity. However, the panel did not recommend the use of these scores in clinical practice. The clinical items deemed to be appropriate for disease assessment included stool frequency and rectal bleeding, which are symptom-based items of the UCDAI and the MCS. For the MCS, a stool frequency score of 2 and a rectal bleeding score of 1 were deemed to be appropriate for defining mild-to-moderate UC disease activity. The presence of fecal urgency should be used to assess mild-to-moderate UC, on a global rating scale. The presence of UC-related fever ruled out mild-to-moderate disease activity.

### 3.3. Endoscopy-Based Disease Activity Assessments

Endoscopic measurements were judged to be important for defining mild-to-moderate UC. The MCS, the Modified MCS (MMCS) or the Ulcerative Colitis Endoscopic Index of Severity (UCEIS) should be used to assess mild-to-moderate UC disease activity. Endoscopic items deemed to be appropriate for disease measurement included the appearance of the mucosa, the vascular pattern, bleeding and erosion/ulcers. When evaluating UC activity endoscopically, the panel determined that a Mayo Endoscopic Sub-score (MES) of 1 is appropriate for mild-to-moderate disease. For each UCEIS item, the panel determined that the patchy obliteration of vascular patterns, mucosal bleeding and the presence of erosions were appropriate for defining mild-to-moderate UC disease activity.

### 3.4. Histology-Based Disease Activity Assessments

The panel did not agree on histology-based measurements, and so, these should not be included in the definition of mild-to-moderate UC. There is a lack of robust evidence concerning the putative association between the histological grade and disease activity.

### 3.5. Biomarker-Based Disease Activity Assessments

The panel considered that biomarker-based measurements were important for defining mild-to-moderate UC. Fecal calprotectin was considered to be an appropriate marker for classifying disease activity in mild-to-moderate UC. However, the panel could not agree on whether a minimum fecal calprotectin cut-off should be applied to inpatients with mild-to-moderate disease. Furthermore, there was uncertainty as to whether CRP levels are an appropriate marker for classifying disease activity in mild-to-moderate UC as CRP can be correlated with disease severity.

### 3.6. Composite Disease Activity Scales

The panel agreed that an MCS score of at least 4 (including an endoscopic sub-score of at least 2 and a rectal bleeding sub-score of at least 1) should be used to define mild-to-moderate UC disease activity. The experts emphasized that an MCS score cut-off had not been fully validated in the assessment of UC disease activity.

### 3.7. Quality of Life/Disability-Based Definitions of Mild-to-Moderate UC

The panel agreed that quality of life, disability, fatigue and work productivity measurements were important when defining mild-to-moderate UC; the latter should not have more than a small impact on daily activities.

## 4. Discussion

For RCTs in patients with UC, the lack of a commonly accepted definition of mild-to-moderate disease activity means that the data are heterogeneous and poorly replicable. This can have a negative impact in clinical practice via the undertreatment or overtreatment of patients. Our international panel of experts suggested a consensus list of items that should be included in the definition of mild-to-moderate disease activity in adult patients screened for inclusion in UC clinical trials (Table 2).

Using a modified RAND/UCLA method, a consensus was reached for 29 statements. The expert panel members came from different countries/continents and diverse practice settings. However, not all countries with expertise in the field of IBD were represented in this panel of experts. The consensus statements were considered for use in future RCTs in patients with UC.

The panel members emphasized the need to combine clinic and endoscopic evaluations when seeking to define mild-to-moderate UC. They agreed that fecal urgency, stool frequency and rectal bleeding are appropriate in the assessment of mild-to-moderate UC disease activity. Fecal urgency is one of the common and most disabling symptoms that patients with UC experience [14,15,16,17]. Although fecal urgency is a key symptom for defining severity of disease activity in clinical practice and has a particularly distressing impact on patients, it is not included in the tools currently used to define IBD severity [18]. The Urgency Numeric Rating Scale (NRS), a validated score to evaluate severity of fecal urgency in adult patients with UC, could be used for a more appropriate and extensive evaluation and categorization of disease activity [19].

The panel members agreed that it is appropriate to use the endoscopic-based items of the UCEIS and the MCS to assess mild-to-moderate UC disease activity: vascular pattern, bleeding, erosions and ulcers. Several composite disease activity scales were considered by the panel. An MCS score of at least 4 (including an endoscopic sub-score of at least 2 and a rectal bleeding sub-score of at least 1) was voted as being appropriate; this decision is consistent with the results of a recent study [10]. Sedano et al. defined mild-to-moderate active UC on the basis of a MCS of 4 to 9 and an MES ≥2 combined with a Rectal Bleeding Sub-score (RBS) ≥1, and a Stool Frequency Sub-score ≥1 or MES ≥1 and a Geboes score > 2.0 or Robarts Histopathology Index (RHI) ≥10 and/or fecal calprotectin > 250 µg/g [10]. In our consensus, there was no agreement on the use of histological measurements to define mild-to-moderate UC. The panel members determined that fecal calprotectin is an appropriate marker for classifying disease activity in mild-to-moderate UC. However, an optimal fecal calprotectin cut-off has not yet been determined [20]. Calprotectin levels are not correlated with disease activity and can be affected by disease extension and blood in the stool samples. Significant intraindividual variations are seen.

In line with the guidelines issued by regulatory authorities, the panel considered that it was inappropriate to use the subjective PGA sub-score to assess disease activity [11]. The PGA sub-score is not derived directly from the patient and cannot adequately determine whether or not major symptoms are relieved [11].

UC can have a major impact on a patient’s life [21]. The panel agreed that mild-to-moderate UC should be defined as disease that does not have more than a small impact on the patient’s daily activities.

The present study provided a consensus definition of mild-to-moderate disease activity in UC. The definition could be used as an inclusion criterion in RCTs in the field of UC. With the ultimate goal of improving patient care and quality of life, there is a constant need for therapeutic trials in patients with mild-to-moderate UC. The results of our initiative will lead to higher-quality clinical studies in mild-to-moderate UC and will facilitate comparison of the latter’s results.

## Figures and Tables

**Table 1 medicina-59-00183-t001:** Approved statements for definition of mild-to-moderate ulcerative colitis for inclusion in clinical trials.

Proposed Statements	Median Panel Score
Symptom measurements are important to define mild-to-moderate ulcerative colitis.	8
The symptom-based items of the UCDAI should be used to assess mild-to-moderate ulcerative colitis disease activity	7
The symptom-based items of the MCS should be used to assess mild-to-moderate ulcerative colitis disease activity.	8
Stool frequency should be used to assess mild-to-moderate ulcerative colitis disease activity.	8
MCS stool frequency score of 2 is appropriate for defining mild-to-moderate ulcerative colitis disease activity	7
Rectal bleeding should be used to assess mild-to-moderate ulcerative colitis disease activity.	8
MCS rectal bleeding score of 1 is appropriate for defining mild-to-moderate ulcerative colitis disease activity.	8
The presence of fecal urgency should be used to assess mild-to-moderate ulcerative colitis disease activity.	8
Fecal urgency should be defined according to a global rating scale	7
Endoscopic measurements are important to define mild-to-moderate ulcerative colitis.	9
The endoscopic-based items of the MCS should be used to assess mild-to-moderate ulcerative colitis disease activity.	8
The endoscopic-based items of the MMCS should be used to assess mild-to-moderate ulcerative colitis disease activity.	7
The endoscopic-based items of the UCEIS should be used to assess mild-to-moderate ulcerative colitis disease activity.	7
Mucosal appearance should be used to assess mild-to-moderate ulcerative colitis disease activity.	7
MES score of 1 for mucosal appearance based on the MCS is appropriate for defining mild-to-moderate ulcerative colitis disease activity.	8
Vascular pattern should be used to assess mild-to-moderate ulcerative colitis disease activity	7
Patchy obliteration of vascular pattern based on the UCEIS is appropriate for defining mild-to-moderate ulcerative colitis disease activity.	7
Bleeding should be used to assess mild-to-moderate ulcerative colitis disease activity.	8
Mucosal bleeding based on the UCEIS is appropriate for defining mild-to-moderate ulcerative colitis disease activity.	7
Erosions and ulcers should be used to assess mild-to-moderate ulcerative colitis disease activity.	7
The presence of erosions based on the UCEIS are appropriate for defining mild-to-moderate ulcerative colitis disease activity.	7
Biomarker measurements are important to define mild-to-moderate ulcerative colitis.	7
The fecal calprotectin level is an appropriate marker for classifying disease activity in mild-to-moderate ulcerative colitis.	8
MCS score of at least 4 including an endoscopic sub-score of at least 2 and a rectal bleeding sub-score of at least 1 should be used to define mild-to-moderate ulcerative colitis disease activity.	7
Quality of life-based measurements are important to define mild-to-moderate ulcerative colitis.	7
Disability based measurements are important to define mild-to-moderate ulcerative colitis.	7
Fatigue measurements are important to define mild-to-moderate ulcerative colitis.	7
Work productivity measurements are important to define mild-to-moderate ulcerative colitis.	7
Mild-to-moderate ulcerative colitis should be defined as disease that does not have a significant impact on daily activities.	7

UCDAI: ulcerative colitis disease activity index; MCS: Mayo clinic score; MMCS: modified Mayo clinic score; UCEIS: Ucerative Colitis Endoscopic Index of Severity; MES: Mayo endoscopic score.

**Table 2 medicina-59-00183-t002:** Proposal for a definition of mild to moderate ulcerative colitis for clinical trials.

Mayo Clinic score of at least 4 including: ▪ Endoscopic sub-score of at least 2 ▪ Rectal bleeding sub-score of at least 1 ▪ No significant impact on the patient’s daily activities

## Data Availability

The data underlying this article are available in the article and in its online Appendix A.

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
