# Peer review of "International Consensus on Definition of Mild-to-Moderate Ulcerative Colitis Disease Activity in Adult Patients"

_medicina, 2023, doi:10.3390/medicina59010183_

Round 1

Reviewer 1 Report

In this review article by Caron et al. the issue of mild to moderate ulcerative colitis is discussed with international consensus.The focus of the paper is well phrased. It is scientifically sound and contains sufficient interest and originality to merit publication. I think this paper is suitable for publication in Journal of Medicina.

Author Response

In this review article by Caron et al. the issue of mild to moderate ulcerative colitis is discussed with international consensus.The focus of the paper is well phrased. It is scientifically sound and contains sufficient interest and originality to merit publication. I think this paper is suitable for publication in Journal of Medicina.

A: We sincerely thank reviewer 1 for her/his feedback and this positive comment.

Reviewer 2 Report

This is a review of the international consensus on disease activity in mild to moderate ulcerative colitis. Twelve physicians from around the world participated in a virtual consensus meeting for discussion and voting. There are various criteria for evaluating mild and moderate activity, and it is significant that an international consensus was reached at this meeting. It is expected that this consensus will be widely referenced in clinical research and other applications in the future.

Author Response

Reviewer: 2

This is a review of the international consensus on disease activity in mild to moderate ulcerative colitis. Twelve physicians from around the world participated in a virtual consensus meeting for discussion and voting. There are various criteria for evaluating mild and moderate activity, and it is significant that an international consensus was reached at this meeting. It is expected that this consensus will be widely referenced in clinical research and other applications in the future.

A: We sincerely thank reviewer 2 for her/his feedback and this positive comment.

Reviewer 3 Report

I see some great effort in standardizing the definition of mild to moderate UC, which is very important to the field. However, I hope the article can have a more clear and strong conclusion. The conclusion should be a finalized summary of all the criteria and cutoffs that the expert panel decided. The conclusion may be in the format of a table (text may also work), for future RCT designers to follow easily and unambiguously. This conclusion should include all the Score systems and biomarker that are recommended to use by this study, and their cutoffs. In the current version of article, the closest thing to this conclusion is the table 1. However, table 1 lacks some key components for future RCT designer’s to follow clearly and unambiguously. Here are three examples.

1)      “The symptom-based items of the UCDAI should be used to assess mild to moderate ulcerative colitis disease activity” What are those items, and how should people use them to define mild to moderate UC? In other words, what are the items and their cutoffs, respectively?

2)      “Stool frequency should be used to assess mild to moderate ulcerative colitis disease activity”

What is the exact range of this frequency in order to define mild to moderate UC?

3)      “Rectal bleeding should be used to assess mild to moderate ulcerative colitis disease activity”

What degree or severity of rectal bleeding is recommended to define this?

USA contributes to a big portion of UC patients. So pls justify why no clinical expert from US was included.

Minor comments:

Line 119: it may be better to change “Histological based” into “histology-based”

Line 159 “the panel members could not agree on whether C-reactive protein is an appropriate marker for classifying disease activity in mild-to-moderate UC because it can be correlated with disease severity.” I don’t think this statement makes sense to me. If CRP correlates with disease severity, it seems to me that makes CRP a perfect marker to define “mild to moderate” disease (severity). Maybe the authors wanted to convey that the range of CRP corresponding to mild to moderate UC is still controversial.

Line 209 subscore not is not

Author Response

Reviewer: 3

I see some great effort in standardizing the definition of mild to moderate UC, which is very important to the field.

A: We sincerely thank reviewer 3 for her/his feedback and this positive comment.

However, I hope the article can have a more clear and strong conclusion. The conclusion should be a finalized summary of all the criteria and cutoffs that the expert panel decided. The conclusion may be in the format of a table (text may also work), for future RCT designers to follow easily and unambiguously. This conclusion should include all the Score systems and biomarker that are recommended to use by this study, and their cutoffs. In the current version of article, the closest thing to this conclusion is the table 1.

A: We thank reviewer 3 for this valuable suggestion. The manuscript was adapted accordingly.

However, table 1 lacks some key components for future RCT designer’s to follow clearly and unambiguously. Here are three examples.

1)      “The symptom-based items of the UCDAI should be used to assess mild to moderate ulcerative colitis disease activity” What are those items, and how should people use them to define mild to moderate UC? In other words, what are the items and their cutoffs, respectively?

A: The clinical items deemed to be appropriate for disease assessment included stool frequency and rectal bleeding which are symptom-based items of the UCDAI and the MCS. For the MCS, a stool frequency score of 2 and a rectal bleeding score of 1 were deemed to be appropriate for defining mild-to-moderate UC disease activity. We used the cutoffs of the MCS which are similar to those of the UCDAI. The manuscript was adapted accordingly.

2)      “Stool frequency should be used to assess mild to moderate ulcerative colitis disease activity”

What is the exact range of this frequency in order to define mild to moderate UC?

A: For the MCS, a stool frequency score of 2 was deemed to be appropriate for defining mild-to-moderate UC disease activity. We used the cutoffs of the MCS which are similar to those of the UCDAI.

3)      “Rectal bleeding should be used to assess mild to moderate ulcerative colitis disease activity”

What degree or severity of rectal bleeding is recommended to define this?

 A: For the MCS, a rectal bleeding score of 1 were deemed to be appropriate for defining mild-to-moderate UC disease activity. We used the cutoffs of the MCS which are similar to those of the UCDAI.

USA contributes to a big portion of UC patients. So pls justify why no clinical expert from US was included.

A: This limit was added in the discussion.

Minor comments:

Line 119: it may be better to change “Histological based” into “histology-based”

A: We thank reviewer 3 for this valuable suggestion. The manuscript was adapted accordingly.

Line 159 “the panel members could not agree on whether C-reactive protein is an appropriate marker for classifying disease activity in mild-to-moderate UC because it can be correlated with disease severity.” I don’t think this statement makes sense to me. If CRP correlates with disease severity, it seems to me that makes CRP a perfect marker to define “mild to moderate” disease (severity). Maybe the authors wanted to convey that the range of CRP corresponding to mild to moderate UC is still controversial.

A: For the experts, there was uncertainty if CRP levels are an appropriate marker for classifying disease activity in mild to moderate UC as CRP can be correlated with disease severity. The manuscript was adapted accordingly.

Line 209 subscore not is not

A: We thank reviewer 3 for this valuable suggestion. The manuscript was adapted accordingly.